# The Effects of Ventilation, Humidity, and Temperature on Bacterial Growth and Bacterial Genera Distribution

**DOI:** 10.3390/ijerph192215345

**Published:** 2022-11-20

**Authors:** Yujia Qiu, Yan Zhou, Yanfen Chang, Xinyue Liang, Hui Zhang, Xiaorui Lin, Ke Qing, Xiaojie Zhou, Ziqiang Luo

**Affiliations:** 1Department of Physiology, The School of Basic Medicine Science, Central South University, Changsha 410000, China; 2Center for the Built Environment, University of California at Berkeley, Berkeley, CA 2506, USA; 3China Vanke Co., Ltd., Changsha 410000, China; 4School of Environmental and Municipal Engineering, Qingdao University of Technology, Qingdao 266061, China

**Keywords:** humidity, temperature, ventilation, air velocity, bacteria, DNA sequencing

## Abstract

Background: Bacteria are readily nourished in airtight environments with high humidity, such as storage cabinets, clothing closets, and corners, where ventilation is normally low and humidity is high. Objectives: We characterized the role of humidity and ventilation in bacterial growth and genus distribution at different temperatures (26 °C and 34 °C). Methods: Fresh pork, which was used as the substrate for bacterial culture, was placed in storage cabinets. Bacterial growth and genera distribution on the surface of pork placed in a storage cabinet under different temperatures (26 °C and 34 °C); relative humidity levels (RH: 50%, 70%, 90%); and ventilation conditions (no ventilation and low, medium, and high levels of ventilation) were assessed by rDNA sequencing. Results: Increased ventilation and reduced humidity significantly decreased bacterial growth at 26 °C and 34 °C. The contribution of increased ventilation to the reduction in bacterial growth exceeded that of decreased humidity. Ventilation had the greatest effect on reducing bacterial growth compared to the unventilated conditions at 70% RH. At 34 °C, medium and high levels of ventilation were required to reduce bacterial growth. High temperatures greatly increased bacterial growth, but ventilation could reduce the degree of this increase.

## 1. Introduction

Bacterial growth and survival depend on many parameters related to the bacterial strain and environment (e.g., temperature and relative humidity). Adequate nutrients; energy; and a suitable environment, including temperature, pH, gas conditions, and osmotic pressure, are necessary for the growth and reproduction of bacteria [1,2]. Moreover, the presence of water is a necessary condition for bacterial reproduction, and bacteria often grow in damp environments [3]. In a suitable temperature range, available water becomes a necessary component and is an important condition for bacteria survival. Bacteria cannot grow or reproduce normally in dry environments.

Storage cabinets are common in households and are used to store household items. To maintain the tidiness of a kitchen, many people store dishes, chopsticks, knives, and cutting boards in storage cabinets, which poses a risk of bacterial infection because of the high humidity inside such cabinets [4,5,6]. Storage cabinets are typically enclosed, with little ventilative air exchange, very low interior air movement, and often high levels of humidity. Bacteria in cabinets are important sources of infection and pollution and have adverse effects on air quality and human health, sometimes leading to infections of the respiratory and digestive systems [7,8]. Additionally, spaces such as clothing closets, where air is tight and humidity is often high, may also be associated with enhanced bacterial growth [9].

With the improvements in the quality of life, public awareness regarding health issues has strengthened significantly. Appropriate ventilation and the maintenance of lower temperatures and humidity levels can be used to control microorganism infections. The relationship between temperature, humidity, and bacteria is complex [10], and controlling the ambient temperature and humidity may not achieve the expected protective effect. Additionally, air conditioning is required, which is energy-intensive. This control is not always possible, for example, in naturally ventilated buildings, where air conditioning is not available. Ventilation, on the other hand, can be easily achieved using fans [11], making this a low-cost and energy-efficient solution for all populations and building types. In our study, we investigated to the effect of temperature, humidity, and ventilation on the growth of bacteria in storage cabinets and identified the most energy-efficient methods for reducing bacterial growth.

The objective of this study was to quantify the impact of humidity and ventilation on the bacterial growth and genus distribution in a cabinet at 26 °C and 34 °C, which represent the average spring and summer temperatures, respectively, in the southern region of China. Fresh pork, which is recognized as one of the most perishable foods [12], was placed in a storage cabinet and used as the test substrate. We assessed the bacterial growth and genera distribution on the surface of pork stored in a cabinet under different temperature, humidity, and ventilation conditions.

## 2. Materials and Methods

### 2.1. Experiment

The experiment was performed in the Enthalpy Difference Laboratory of Dongda Air Conditioning, Yuyao, Zhejiang Province, China. The test chamber dimensions were 5 m (length) × 4.2 m (width) × 3.5 m (height), and the temperature and humidity could be controlled. We established four test cabinets in the chamber to represent four ventilation levels. Details of the cabinet (size: 0.8 m (length) × 0.75 m (width) × 0.6 m (height)) are provided in Appendix A. Each cabinet was divided into two compartments (photos in Figure 1A,B show the front and side views). The supply compartment (left-hand side shown in Figure 1D, 0.3 m width) contained a low-power fan, which drew unfiltered air from the chamber through a supply vent to the supply compartment at different flow rates. The other compartment was the test compartment (right-hand side shown in Figure 1D, 0.45 m width). There was a perforated divider between the two compartments, with many small, evenly distributed holes (grey dots in Figure 1C). The fan created positive pressure in the supply compartment, which pushed air evenly through the small holes in the perforated divider to the test compartment. The ventilation outlets (two outlets with a size of 0.3 m (length) × 0.03 m (height)) were located on opposite sides of the divider of the test compartment connected to the chamber, 0.17 m below the ceiling of the compartment (shown in black in Figure 1A and green in Figure 1C,D).

Six conditions were tested in the chamber: temperatures of 26 and 34 °C, and relative humidity levels of 50, 70, and 90%. Under each set of environmental conditions, zero-, low-, medium-, and high-ventilation levels were established for each of the four test cabinets. Assuming plug flow across the test compartment from the perforated supply divider to near the exhaust wall, the velocities adjacent to the meat sample were 0.02, 0.06, and 0.1 m/s (herein termed cross-section velocity). The velocity in the unventilated cabinet was 0.0 m/s; there was no supply inlet and no ventilation outlet. The corresponding four velocities at the ventilation outlets were 0, 0.56, 1.67, and 2.78 m/s. The air speed at each ventilation outlet of the cabinet was measured using an anemometer at four evenly distributed points across the outlet to make sure that the air velocity was uniformly spread. Four pieces of pork were placed in each test compartment, in four petri dishes on top of a shelf (Figure 1C–E). The pork was exposed to different working conditions 24 h before bacteria were removed.

### 2.2. Equipment

An electronic temperature and humidity data logger DS1923 (Wadisen Electronic Technology Co., Ltd., Shanghai, China) was used in the experimental cabinet to record real-time temperature and humidity every minute. Wind speed was measured using a 9535/3545 TSI air volume and flow rate meter (Connor VELOCICALC, Minneapolis, MN, USA). All operating instruments were sterilized by autoclaving or ultraviolet irradiation before use. Bacterial counts were performed inside sterile ultra-clean workbenches. Culture agar, sterile petri dishes, and sterile cell scrapers were purchased from Sheng gong Biological Company (Shanghai, China).

### 2.3. Meat and Agar Preparation

During pilot tests, we found that diversified bacteria grew quickly on fresh pork. Therefore, to better explore the effects of temperature, humidity, and ventilation on bacterial growth, fresh pork was chosen as the test substrate. Fresh pork was purchased from a local market and washed with sterile water in the laboratory within 0.5 h of purchase. The pork was cut into pieces of equal size (approximately 1 cm × 1 cm × 1 cm and 2 g in weight). An amount of 23.5 g of agar powder (Solarbio, P9270, Beijing, China) was dissolved in 1000 mL distilled water and autoclaved at 121 °C for 15 min. It was then kept in a water bath at 45 °C until use.

### 2.4. Bacteria Collection and Calculation

After cultivation for 24 h under different conditions in the four testing cabinets, the pork pieces were soaked and washed in 5 mL sterile phosphate-buffered saline (PBS) and the solution was collected (containing bacteria from the surface of the cultured pork). following the plate count method [13,14,15,16], the bacterial liquid collected from the meat was inoculated into a sterile tube. Then, 1mL of the sample stock solution was transferred into a sterile tube containing 9 mL of PBS, shaken, and mixed evenly to prepare a 1:10 sample homogenate. This was repeated once for each incremental dilution (1:100, 1:1000). To estimate sample contamination, one-to-three appropriate dilution concentrations were selected, and 1 mL of the diluted sample was transferred to a sterile Petri dish. We poured 15–20 mL of plate-counting agar, which cooled to 46–50 °C, into the prepared Petri dish and immediately mixed it with the diluted sample. Following agar solidification, the plate was turned and incubated at 37 °C for 48 h. The colony count was expressed as colony-forming units (CFU), with 30–300 colonies being suitable. The dilution multiples and corresponding number of colonies in different Petri dishes were recorded. The total number of bacterial colonies on the Petri dishes × the dilution multiple was used to obtain the total number of bacteria under the test conditions (temperature, humidity, and ventilation). We repeated the procedure using meat pieces that were washed with tap water (without 24 h culture) as the base reference value, with 30–300 colonies being suitable, and then divided the total number of bacteria in the individual pieces under each set of test conditions by the base value to obtain the growth rate. The experimental operation is shown in Figure 2.

### 2.5. High-Throughput 16S rDNA Sequencing

First, we collected 1mL of bacterial liquid from each piece of pork for 16S rDNA sequencing [15,17] to explore the distribution of bacterial genera in each sample. Total genomic DNA was extracted from collected bacterial liquid using a DNeasy PowerSoil Kit (QIAGEN, 12888–100, Germany), following the manufacturer’s instructions. Agarose gel electrophoresis was performed to verify the quality and quantity of DNA. The V3-V4 variable region of bacterial 16S rRNA was amplified with universal primers 343F-5′-TACGGRAGGCAGCAG-3′ and 798R-5′-AGGGTATCTAATCCT-3′. Double V regions can provide more reference amplification sequences and improve the accuracy of genera distribution classification. After amplification, all PCR products were sequenced on the MiSeq Illumina platform at OE Biotech (Shanghai, China) for high-throughput 16S rDNA sequencing. Sequencing analysis followed the standard operating procedures of OE Biotech. Paired-end reads were preprocessed using Trimmomatic software to detect and move ambiguous bases (N). After trimming, paired-end reads were assembled using FLASH software. Non-target sequences were removed, and chimeras were detected using UCHIME software. The representative sequences of each OTU were selected using QIIME software, and all representative sequences were compared and annotated with a database. The Greengenes or Silva (version 123) database was used for 16S comparisons to determine the proportion of different bacterial genera.

After determining the proportion of different bacterial genera in the bacterial solution and multiplying the total number of bacteria by the obtained ratio of different bacterial genera, the actual number of bacterial genera in each sample was obtained.

### 2.6. Statistical Analysis

One-way analysis of variance was used to calculate statistical significance. Differences were considered statistically significant at *p* < 0.05. Statistical tests were performed using GraphPad Prism version 7.00 (GraphPad Software). MATLAB 7.0.1 (MathWorks, Natick, MA, USA) software was used to develop a three-dimensional diagram of the equation.

## 3. Results

### 3.1. Real-Time Temperature and Humidity under Different Working Conditions

The real-time temperature and humidity in the four test cabinets over the 24 h experiment are shown in Figure 3A,B. As shown in Figure 3A, there was no obvious difference in the real-time temperatures between unventilated and ventilated cabinets (26 °C or 34 °C, Figure 3A), and fluctuations were similar in all cabinets (unventilated and ventilated), within 1.0 °C. The results revealed that ventilation had no obvious effect on temperature within the test cabinets. For real-time humidity, the results showed that the humidity in the unventilated cabinet at 50 and 70% RH was slightly higher than that in the ventilated cabinets; at 90% RH, there was no difference. Fluctuations in humidity in the ventilated cabinet were sometimes slightly above and below the humidity in the unventilated cabinet (Figure 3B). These results suggested that ventilation had no obvious effect on the humidity within the test cabinet.

### 3.2. Effect of Humidity and Ventilation on Bacterial Growth at 26 °C

*Unventilated cabinet.* Under the conditions of 90, 70, and 50% RH in the unventilated test cabinet, the total number of bacteria in the meat (compared to the base value of 3.0 × 10^3^, measured after washing in tap water, before placing in the cabinets) increased by 3.06 × 10^4^, 1.38 × 10^4^, and 4.9 × 10^3^ times, respectively, after 24 h culture. Increasing the humidity from 50 to 70% RH and from 50 to 90% RH increased the bacterial growth by 2.8 and 6.2 times, respectively; increasing the humidity from 70 to 90% RH increased the bacterial growth by 2.2 times (Figure 4A and Table 1). This suggested that reducing the humidity at 26 °C reduced bacterial growth.

*Comparison between ventilated cabinets and the unventilated cabinet.* Ventilation significantly reduced bacterial growth under all tested humidity conditions. Under the high-humidity conditions (90%), low, medium, and high levels of ventilation significantly reduced bacterial growth by 9.6, 9.3, and 16.4 times compared to the unventilated cabinet. These reductions were much larger than those observed when the humidity was reduced from 90 to 50% RH in the unventilated cabinet (6.2 times) (Figure 4A and Table 1). Thus, increasing the ventilation speed, even at a low level, was more effective at reducing bacterial growth than reducing humidity.

Ventilation had the greatest impact on reducing the bacterial growth under the 70% RH test conditions (Figure 4B and Table 1). Low, medium, and high levels of ventilation reduced bacterial growth by 14.8, 71.1, 59.0 times, respectively, compared with no ventilation. These reductions were much higher than those observed under the 90% RH test conditions (9.6-, 9.26-, 16.4-fold reductions in bacterial growth compared with no ventilation) and the 50% RH test conditions (11.0-, 9.4-, 10.9-fold reductions in bacterial growth compared with no ventilation). Under the low-humidity (50% RH) conditions, the presence or absence of ventilation had a significant impact, but the differences between the various levels of ventilation were very small, all presenting a roughly 10-fold change in bacterial growth compared with the unventilated cabinet (Figure 4C and Table 1). Taken together, these results showed that: (I) increasing the ventilation was much more effective than reducing the humidity across all tested conditions from 50 to 90% RH, with the most effective results observed under 70% RH; (II) low ventilation levels could make a significant improvement; and (III) increasing the ventilation and reducing the humidity could decrease the bacterial growth at 26 °C.

### 3.3. Effects of Humidity and Ventilation on Bacterial Growth at 34 °C

*Unventilated cabinet.* Under the conditions of 90, 70, and 50% RH at 34 °C in the unventilated test cabinet, the total number of bacteria in the meat increased by 802.5 × 10^4^, 193.3 × 10^4^, and 8.7 × 10^4^ times, respectively, compared to the baseline value after 24 h culture (Figure 5A and Table 2). Temperature had a significant impact on the bacterial growth in the meat. Bacterial growth increased 262.4-, 140.6-, 17.6-fold as the temperature increased from 26 to 34 °C under 90, 70, and 50% RH, respectively. The significant increase at 70 and 90% RH compared with 50% RH at 34 °C indicated that bacterial growth in the meat was fastest under hot and humid conditions, and that reducing the humidity to 50% RH was effective.

At 34 °C, humidity had a greater effect on bacterial growth than at 26 °C, but the increase was not linear. When the humidity increased from 50 to 70% and from 50 to 90%, the bacterial growth increased 22.3- and 92.6-fold. Increasing the humidity from 70 to 90% increased the bacterial growth 4.2-fold.

*Ventilated cabinets compared with unventilated cabinets.* At 34 °C, ventilation had a greater effect on bacterial growth reduction than at 26 °C. Under high-humidity conditions (90%), low, medium, and high levels of ventilation significantly reduced bacterial growth by 4.3, 39.9, and 38.0 times compared to the value in the unventilated cabinet (Figure 5A and Table 2). At 70% RH, ventilation was most effective at reducing bacterial growth; under low, medium, and high levels of ventilation, bacterial growth was reduced 2.3-, 169.3-, 166.9-fold compared with the unventilated conditions. At 50% RH, the effects of ventilation were less pronounced than those at 70 and 90% RH, with an approximate 3–4-fold reduction compared with the values obtained under unventilated conditions. This indicated that at high ambient temperatures, ventilation was particularly effective at reducing bacterial growth, especially under high-humidity conditions. In contrast to the results obtained at 26 °C, under a high ambient temperature, low levels of ventilation were less effective at reducing bacterial growth than lowering the humidity from 90 to 50% RH without ventilation.

The reduction in bacterial growth under medium and high levels of ventilation at 90% RH and 34 °C was about 4 times higher than under low levels of ventilation with humidity reduced from 90 to 70% (see Figure 5A and Table 2). The bacterial growth reduction under medium and high levels of ventilation at 70% RH and 34 °C was about 8 times higher than under low levels of ventilation with humidity reduced from 70 to 50% (see Figure 5B and Table 2). Taken together, these results demonstrated that higher levels of ventilation were more effective at reducing bacterial growth at 34 °C than low levels of ventilation with reduced humidity.

Considering the results obtained for ventilation changes at 26 °C, high temperatures significantly increased bacterial growth. Under medium and high levels of ventilation, bacterial growth increased 50–60-fold at 34 °C compared with 26 °C. However, considering the results obtained without ventilation, wherein bacterial growth increased up to 262 times with an increase in temperature, ventilation was especially effective at reducing the bacterial growth under high temperatures.

### 3.4. Bacterial Growth Model for Humidity and Ventilation

To evaluate the effects of ventilation and humidity on bacteria growth in the cabinet, we obtained an equation with fold change in bacterial growth as the dependent variable and ventilation (in terms of the cross-sectional velocity in the test compartment: 0, 0.02, 0.06, and 0.1 m/s) and humidity as independent variables. We then determined the contribution of independent variables to the dependent variable according to the coefficients of the two independent variables under 26 and 34 °C conditions.

At 26 °C, the fitting equation was as follows: bacterial growth (fold change) = e ^(4.549 × humidity − 24.598 × air velocity + 5.159)^. The regression coefficients in the equation indicated that reducing the humidity by 10% or increasing the cross-sectional air velocity by 0.02 m/s exerted a similar effect on reducing bacterial growth. The bacterial growth model at 26 °C is presented in Figure 6A. At 34 °C, the fitting equation was as follows: bacterial growth = e ^(7.434 × humidity − 37.471 × air velocity + 8.364)^. Again, the regression coefficients in the equation indicated that reducing the humidity by 10% or increasing the cross-sectional air velocity by 0.02 m/s exerted a similar effect on bacterial growth. The coefficients for cross-sectional air velocity were 4.549 at 26 °C and 7.434 at 34 °C, and coefficients for humidity were 24.598 and 37.471, respectively, indicating that the contribution of increased ventilation to reducing bacterial growth was greater than that of decreasing the humidity. The bacterial growth model at 34 °C is presented in Figure 6B.

### 3.5. The Effect of Humidity and Ventilation on the Distribution of Bacterial Genera at 26 °C

*Unventilated cabinet.* As shown in Figure 7A, at 26 °C and without ventilation, lowering the humidity from 90 to 70 and 50% RH reduced the numbers of nine bacterial genera, namely *Acinetobacter*, *Myroides*, *Citrobacter*, *Kurthia*, *Serratia*, *Enterobacter*, *Wohlfahrtiimonas*, *morganella*, and *proteus*. However, the differences between the 70 and 50% RH conditions were not statistically significant.

*Ventilated cabinets.* Under the conditions of 90% RH at 26 °C, low, medium, and high levels of ventilation significantly reduced the number of 16 bacteria genera, namely *Acinetobacter*, *Klebsiella*, *Myroides*, *Citrobacter*, *Kurthia*, *Serratia*, *Enterobacter*, *Wohlfahrtiimonas*, *morganella*, *proteus*, *Lactococcus*, *Macrococcus*, *Providencia*, *Escherichia-Shigella*, *Pseudomonas*, and *Vibrio* (Figure 7B). Under the conditions of 70% RH at 26 °C, low, medium, and high levels of ventilation significantly reduced the number of 11 bacteria genera, namely *Acinetobacter*, *Klebsiella*, *Kurthia*, *Serratia*, *Enterobacter*, *Staphylococcus*, *Lactococcus*, *Macrococcus*, *Pseudomonas*, and *Clostridium_sensu_stricto_1* (Figure 7C). Under the conditions of 50% RH at 26 °C, low, medium, and high levels of ventilation significantly reduced the number of eight bacterial genera, namely *Acinetobacter*, *Klebsiella*, *Pseudomonas*, *Sphingobacterium*, *Chryseobacterium*, *Exiguobacterium*, *Delftia*, and *Bacillus* (Figure 7D). As the humidity increased, the ventilation had a stronger effect on reducing bacterial genera; the higher the humidity, the more bacterial genera were reduced via ventilation.

### 3.6. The effect of Humidity and Ventilation on the Distribution of Bacteria Genera at 34 °C

*Unventilated cabinet.* As shown in Figure 8A, at 34 °C and without ventilation, lowering the humidity from 90 to 70 and 50% RH reduced the number of 18 bacteria genera, namely *Bacteroides*, *Prevotella_9*, *Lactobacillus*, *Bifidobacterium*, *Lachnospiraceae_NK4A136_group*, *Faecalibacterium*, *Alloprevotella*, *Alistipes*, *Aeromonas*, *Agathobacter*, *Ruminococcaceae_UCG_014*, *Parabacteroides*, *Helicobacter*, *Ruminococcus_1*, *Prevotellaceae_NK3B31_group*, *Streptococcus*, *Eubacterium_coprostanoligenes_group*, and *Cetobacterium*. However, no difference was observed between the 70 and 50% RH conditions.

*Ventilated cabinets.* Under the conditions of 90% RH at 34 °C, ventilation significantly reduced the number of 21 bacterial genera, namely *Escherichia-Shigella*, *Bacteroides*, *Prevotella_9*, *Lactobacillus*, *Bifidobacterium*, *Macrococcus*, *Lachnospiraceae_NK4A136_group*, *Faecalibacterium*, *Lactococcus*, *Alloprevotella*, *Collinsella*, *Ruminococcaceae_UCG_014*, *Alistipes*, *Agathobacter*, *Parabacteroides*, *Streptococcus*, *Ruminococcus_1*, *Prevotellaceae_NK3B31_group*, *Cetobacterium*, *Helicobacter*, and *Eubacterium_coprostanoligenes_group* (Figure 8B). Under the conditions of 70% RH, low, medium, and high levels of ventilation significantly reduced the number of 21 bacterial genera, namely *Acinetobacter*, *Escherichia-Shigella*, *Bacteroides*, *Klebsiella*, *Prevotella_9*, *Bifidobacterium*, *Lachnospiraceae_NK4A136_group*, *Alistipes*, *Faecalibacterium*, *Collinsella*, *Aeromonas*, *Alloprevotella*, *Agathobacter*, *Ruminococcaceae_UCG_014*, *Macrococcus*, *Helicobacter*, *Ruminococcus_1*, *Prevotellaceae_UCG_001*, *Streptococcus*, *Citrobacter*, *Eubacterium_coprostanoligenes_group*, and *Roseburi* (Figure 8C). Under the conditions of 50% RH, low, medium, and high levels of ventilation significantly reduced the number of Kurthia (Figure 8D). Again, the results demonstrated that the higher the humidity, the greater the effect of ventilation.

## 4. Discussion

RH influences the growth and reproduction of microorganisms. The water content of microbial cells is generally 70–90% [18]. As the RH equilibrium decreases, microbial metabolism is progressively inhibited until the conditions become unsuitable for growth [19]. When humidity is high, a biofilm is formed on the surface of an object; microorganisms live in this biofilm, which provides protection and allows bacteria to survive and thrive in hostile environments [20]. When the air is dry, microorganisms lose water and shrink, and microbial replication stops when the RH equilibrium is below 60% [19]. When the RH increases, microbial activity increases and vice versa. RH affects spores [18] and the length of the growth stagnation period of microorganisms [3,21,22], and exerts different effects on different microorganisms. Our results confirmed that reducing humidity could decrease bacterial growth at 26 and 34 °C. Placing charcoal or a dehumidification bag inside a cabinet can reduce the humidity to a certain degree. In addition, the use of air-conditioning and dehumidification appliances can reduce the humidity in cabinets to some extent. However, the energy consumption of air-conditioning and dehumidification appliances (300–500 w) is high; therefore, the efficiency of reducing humidity in cabinets is not high. These are expensive and inefficient options.

Ventilation and air movement can reduce bacterial growth because moisture evaporation increases significantly as air movement increases, which dries the surfaces of substrates and inhibits bacterial growth. The evaporative heat transfer coefficient increases exponentially as the air movement increases [23]. Based on experience, the growth of microorganisms in cabinets can be reduced by proper ventilation, which can be achieved by opening cabinet doors. However, manually opening doors for ventilation is inconvenient and inefficient. As shown by our results, increasing ventilation from low to high levels using a low-powered fan (power: <20 W) significantly reduced bacterial growth, which is an effective and energy-efficient method. Our results also showed that, at 26 °C, only the low and medium levels of ventilation under 90% RH resulted in a higher bacterial growth reduction than reducing the humidity from 90 to 50% RH. Thus, under high-humidity conditions, increasing the ventilation speed was much more effective at reducing the bacterial growth than reducing the humidity of the air, which even at a low level could effectively reduce the bacterial growth. The effects of ventilation on reducing bacterial growth were stronger than those of reducing the humidity, and it is a more energy-efficient method. The air velocity across the surface of the substrates was not required to be high. Moreover, these findings have great significance for improving the quality of living for individuals with low incomes, because ventilation is affordable, requiring only a fan.

For the ventilated cabinets, the air change per hour (ACH) was quite high, at 36, 108, and 180 times/hour, corresponding to the three ventilation levels, respectively. With a high ACH, we could assume that the bacteria in the air of the three ventilated cabinets were the same as those in the chamber air where the four cabinets were located. With the same level of bacteria in the air of the three ventilated cabinets, we hypothesized that the enhanced reduction in the bacterial growth as the ventilation level increased (from low to medium to high) was mainly caused by the drying of the substrate surfaces with increased air movement.

Our results showed that increasing ventilation and reducing humidity significantly decreased pathogenic bacterial genera. *Acinetobacter* can cause respiratory tract infection, septicemia, meningitis, endocarditis, skin wound infection, and urogenital tract infection [24]. Our results showed that both ventilation and reducing humidity could decrease bacterial growth at 26 °C, and that a proper level of ventilation could further reduce bacterial growth under lower-humidity conditions (50% RH). However, ventilation or reducing the humidity had no significant effect at 34 °C. *Klebsiella* spp. is a conditional pathogen, which can cause pneumonia [25]. Reducing humidity had no effect on the number of *Klebsiella* spp.; however, different levels of ventilation significantly reduced the number of *Klebsiella* spp. *Chryseobacterium* can cause respiratory tract infection and urinary tract infection [26,27]. When the humidity was reduced to 50%, different levels of ventilation could further reduce the numbers of *Chryseobacterium* under 50% RH and 26 °C. In addition, providing ventilation and lowering the humidity could reduce the number of other pathogenic bacterial genera, including *Citrobacter*, to which neonates and other immunocompromised patients are particularly susceptible [28]; *Enterobacter*; *Escherichia-shigella*; and *Myroides*, which causes various infections and death in immunocompromised individuals [29]. Furthermore, different levels of ventilation at 50% RH could further reduce the numbers of many pathogenic bacterial genera, such as *Acinetobacter*, *Klebsiella*, *Pseudomonas*, and *Pseudomonas aeruginosa*, which cause serious infections in immunocompromised patients; *Sphingobacterium; Chryseobacterium; Exiguobacterium* (some of which are related to community-acquired pneumonia and bacteremia [30]); *Delftia*; and *Bacillus* (some of which can cause food intoxication and toxicoinfection [31,32]) at 26 °C and *Kurthia* at 34 °C. Most of the bacteria mentioned here have resistance to antibiotics currently used for the treatment of clinical infections [33,34,35].

Notably, fresh pork was selected as the test substrate in this study, because it supports the rapid growth of bacteria. The use of a substrate that encouraged rapid bacterial growth enabled us to clarify the effects of controlling the temperature, humidity, and ventilation. Future studies could use other foods or household items.

## 5. Conclusions

In summary, our results showed that both increasing the ventilation and reducing the humidity could decrease bacterial growth at 26° and 34 °C. At 26 °C, low ventilation similarly reduced bacterial growth to the levels observed under high ventilation. At 34 °C, medium and high levels of ventilation were needed to effectively reduce bacterial growth.

Ventilation was more effective at reducing bacterial growth under 70% RH than under 90% and 50% RH conditions. A high temperature greatly increased bacterial growth; however, ventilation could help to attenuate this increase. Moreover, both ventilation and reduced humidity affected the distribution of bacterial genera, and ventilation was more effective under higher humidity levels. Under specific humidity conditions, ventilation could reduce the number of multiple bacterial genera, many of which were pathogenic. Thus, ventilation is an effective, energy-efficient, and convenient method for implementation in continuous storage cabinets to inhibit bacterial growth.

## Figures and Tables

**Figure 1 ijerph-19-15345-f001:**
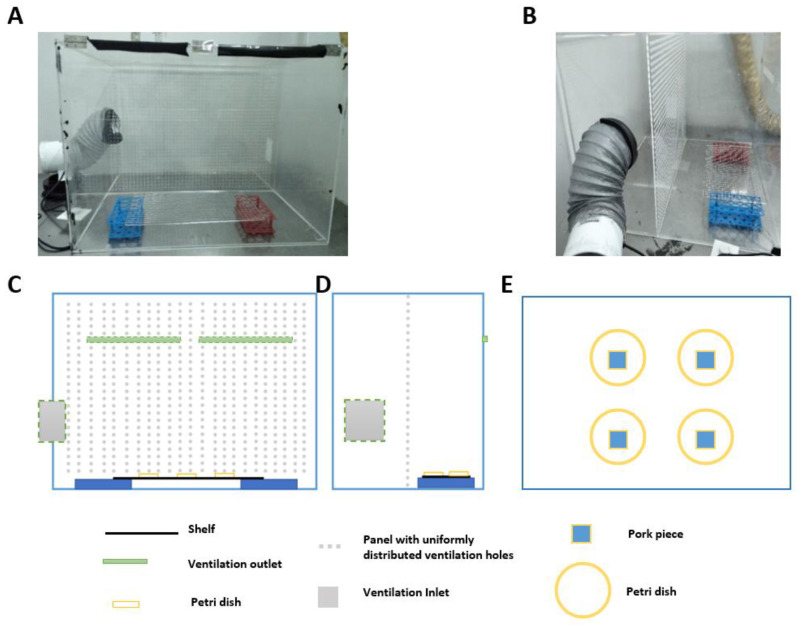
Cabinet working conditions. (**A**) Front view of the experimental cabinet; (**B**) side view of the experimental cabinet; (**C**) front-view diagram of the experimental cabinet; (**D**) side-view diagram of the experimental cabinet; (**E**) floor-view diagram of the experimental cabinet.

**Figure 2 ijerph-19-15345-f002:**
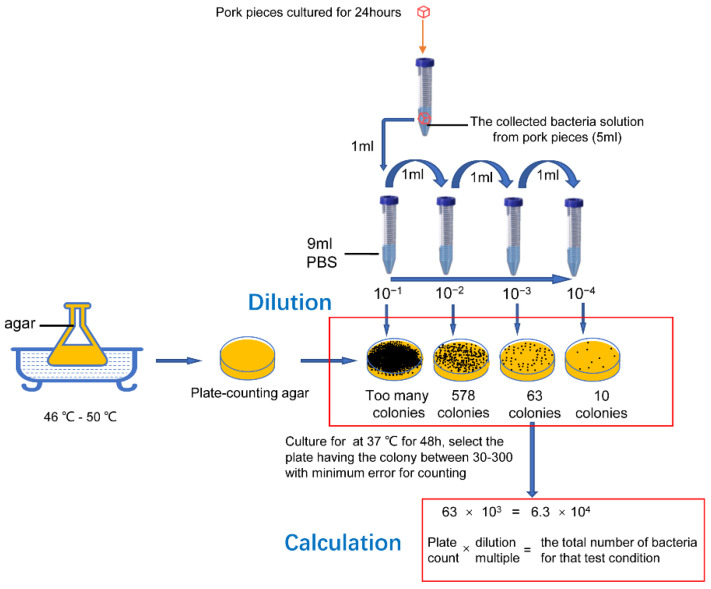
The experimental operation of bacteria collection, cultivation, and counting.

**Figure 3 ijerph-19-15345-f003:**
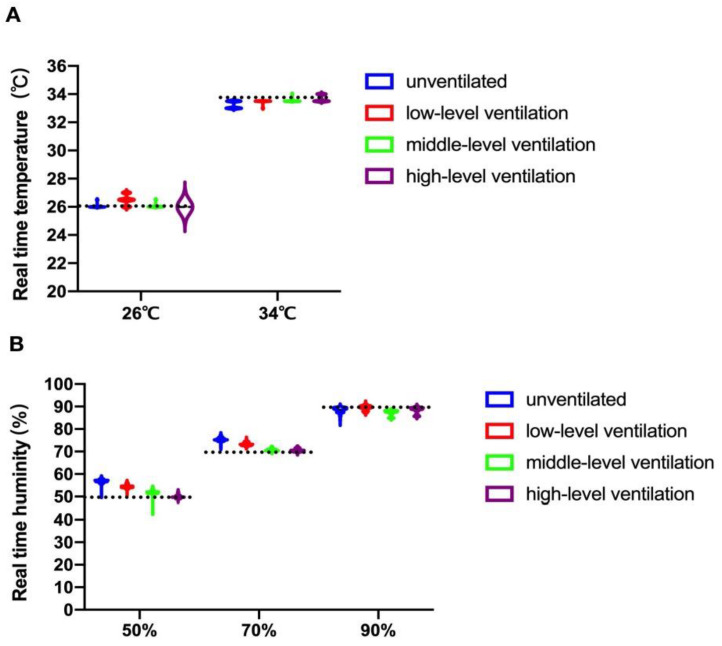
Environmental monitoring of every set of conditions. (**A**) Real-time temperature in each test cabinet with the pre-set chamber temperature of 26 °C or 34 °C; (**B**) Real-time humidity in each test cabinet with the pre-set chamber humidity of 50%, 70%, or 90% RH.

**Figure 4 ijerph-19-15345-f004:**
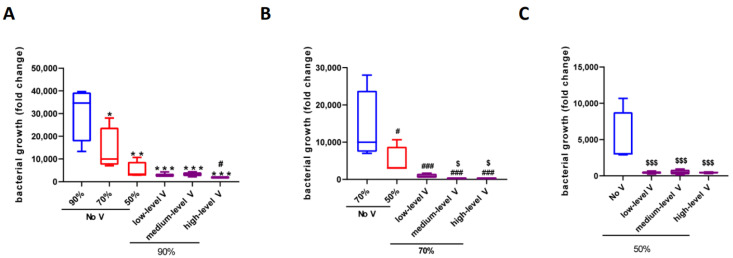
The bacterial growth (fold change compared to the base value measured after washing in tap water, before placing in the cabinets) in meat under different working conditions at 26 °C. (**A**) no ventilation and different ventilation levels at 90%, 26 °C; (**B**) no ventilation and different ventilation levels at 70%, 26 °C; (**C**) no ventilation and different ventilation levels at 50%, 26 °C. * *p* < 0.05 vs. no ventilation 90%; ** *p* < 0.01 vs. no ventilation 90%; *** *p* < 0.001 vs. no ventilation 90%; ^#^ *p* < 0.05 vs. no ventilation 70%; ^###^
*p* < 0.001 vs. no ventilation 70%; ^$^ *p* < 0.05 vs. no ventilation 50%; ^$$$^ *p* < 0.001 vs. no ventilation 50%.

**Figure 5 ijerph-19-15345-f005:**
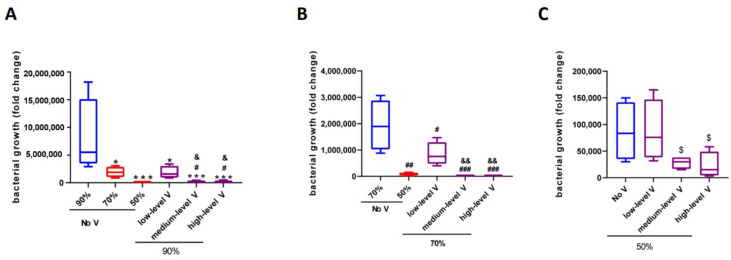
The bacterial growth (fold change compared to the base value measured after washing in tap water, before placing into the cabinets) in the meat under different working conditions at 34 °C. (**A**) No ventilation and different ventilation levels at 90%, 34 °C; (**B**) no ventilation and different ventilation levels at 70%, 34 °C; (**C**) no ventilation and different ventilation levels at 50%, 34 °C. * *p* < 0.05 vs. no ventilation 90%; *** *p* < 0.001 vs. no ventilation 90%; ^#^ *p* < 0.05 vs. no ventilation 70%; ^##^ *p* < 0.01 vs. no ventilation 70%; ^###^
*p* < 0.001 vs. no ventilation 70%; ^$^ *p* < 0.05 vs. no ventilation 50%. ^&^
*p* < 0.05 vs. low-level V; ^&&^
*p* < 0.01 vs. low-level V.

**Figure 6 ijerph-19-15345-f006:**
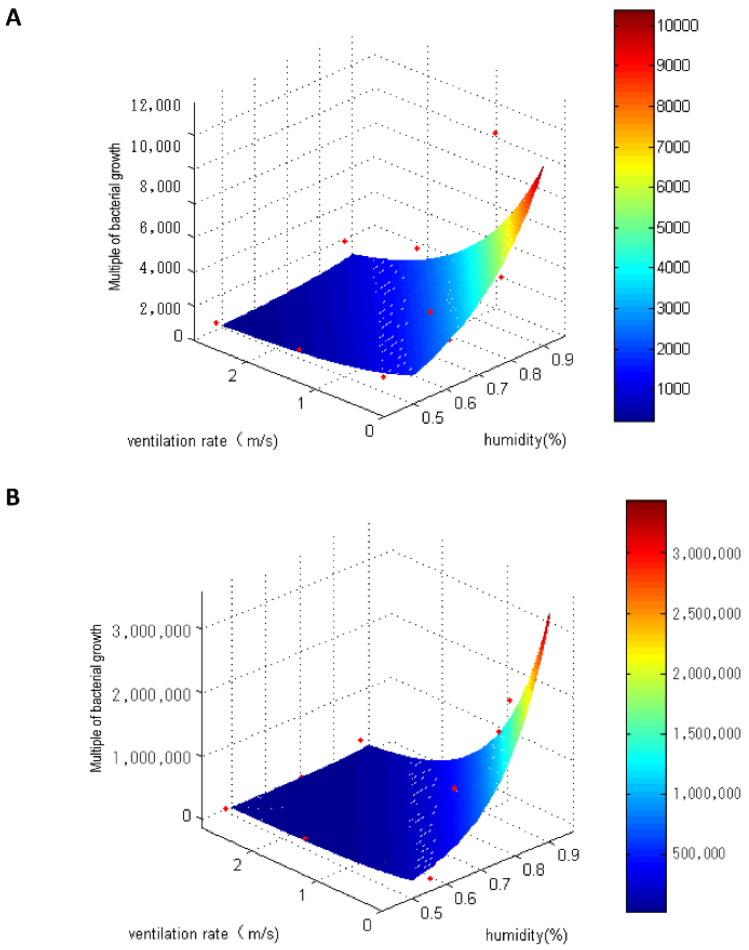
Bacterial growth model for humidity and ventilation. (**A**) at 26 °C; (**B**) at 34 °C. ● represented bacterial growth values in one condition (like 90%, no ventilation).

**Figure 7 ijerph-19-15345-f007:**
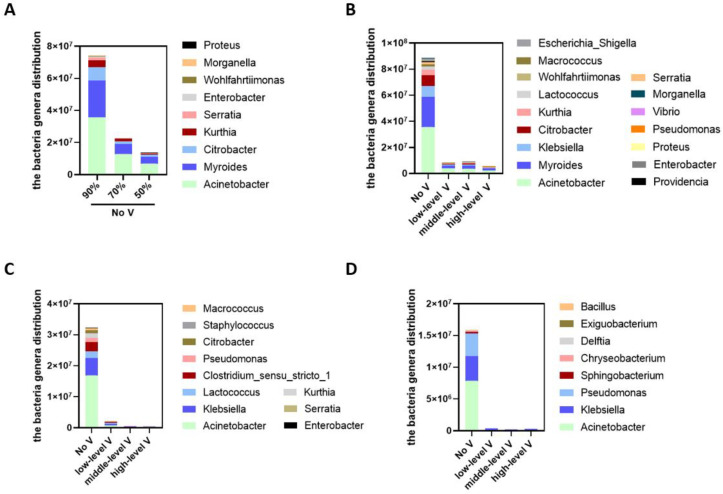
The effect of humidity and ventilation on the bacterial genera distribution at 26 °C. (**A**) The effect of different humidity levels (90%, 70%, 50%) on bacterial genera without ventilation; (**B**) the effect of ventilation on bacterial genera under 90% humidity; (**C**) the effect of ventilation on bacterial genera under 70% humidity; (**D**) the effect of ventilation on bacterial genera under 50% humidity.

**Figure 8 ijerph-19-15345-f008:**
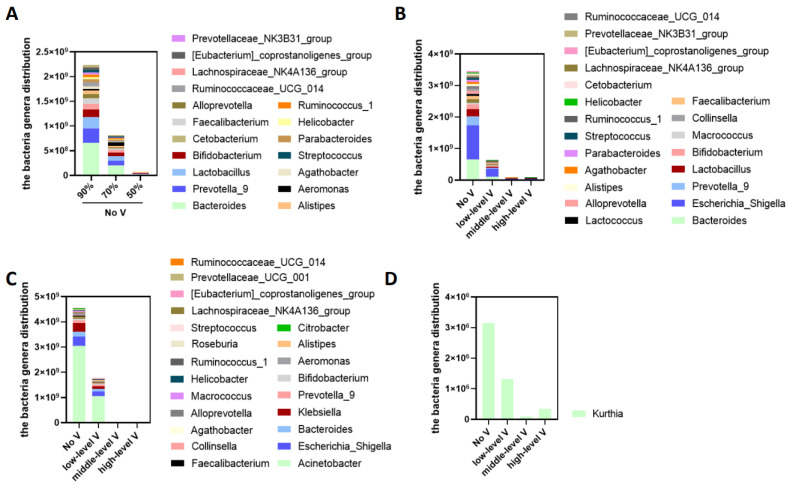
The effect of humidity and ventilation on the bacterial genera distribution at 34 °C. (**A**) The effect of different humidity levels (90%, 70%, 50%) on bacterial genera without ventilation; (**B**) the effect of ventilation on bacterial genera under 90% humidity; (**C**) the effect of ventilation on bacterial genera under 70% humidity; (**D**) the effect of ventilation on bacterial genera under 50% humidity.

**Table 1 ijerph-19-15345-t001:** The bacterial growth (fold change) in meat under different working conditions at 26 °C.

26 °C	No V	Low-Level V	Middle-Level V	High-Level V
90%	3.06 ± 0.61 × 10^4^	3.20 ± 0.56 × 10^3^	3.30 ± 0.47 × 10^3^	1.87 ± 0.08 × 10^3^
70%	1.38 ± 0.48 × 10^4^	9.31 ± 3.18 × 10^2^	1.93 ± 0.33 × 10^2^	2.33 ± 0.47 × 10^2^
50%	4.91 ± 1.92 × 10^3^	4.45 ± 0.95 × 10^2^	5.21 ± 1.77 × 10^2^	4.51 ± 0.49 × 10^2^

mean ± SEM.

**Table 2 ijerph-19-15345-t002:** The bacterial growth (fold change) in meat under different working conditions at 34 °C.

34 °C	No V	Low-Level V	Middle-Level V	High-Level V
90%	8.03 ± 3.44 × 10^6^	1.85 ± 0.56 × 10^6^	2.01 ± 0.78 × 10^5^	2.11 ± 0.97 × 10^5^
70%	1.93 ± 0.48 × 10^6^	8.46 ± 2.24 × 10^5^	1.14 ± 0.13 × 10^4^	1.16 ± 0.20 × 10^4^
50%	8.67 ± 2.81 × 10^4^	8.71 ± 2.89 × 10^4^	2.83 ± 0.59 × 10^4^	2.29 ± 1.24 × 10^4^

mean ± SEM.

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
