# Peer review of "The Effects of Ventilation, Humidity, and Temperature on Bacterial Growth and Bacterial Genera Distribution"

_ijerph, 2022, doi:10.3390/ijerph192215345_

Round 1

Reviewer 2 Report

  In the present manuscript, the authors have analyzed and compared the difference of microbiota in rotted pork under different conditions, such as temperature, humidity, and ventilation.  After refereeing process, I have concerns to publish the submitted manuscript with following insufficient points.

I cannot appreciate the reason why the authors use the pork as a fermentation (rotting) material.  It seems hard to expect the situation that someone leaves a raw pork in storage cabinet, thus the goal (application) of the study is unobvious.

  Further, the “discussion” part seems to contain the repetition of results, and the discussion section may be not satisfactory (too short, and poor in “discussion” itself with appropriate references).

  There are also many typographical errors in the submitted manuscript, therefore the manuscript may not satisfy the criteria for not only the publication but also the submission.

  In addition, there are no descriptions about how perform the microbial analysis.

Round 2

Reviewer 1 Report

Dear authors,

The manuscript was improved, but it still requires a minor revision.

Although the language was enhanced, there are still many parts (especially the ones that were added after the revision) which are not grammatically correct and need revising. Add data about the softwares that you used (version, city, country) throughout the manuscript.

Regarding your explanation about the genera, I suggest you remove or explain the meaning of the numbers after certain genera that you mention (e.g. Ruminococcus_1). As for the Eubacterium coprostanoligenes group, I was pointing out the way you have written the name of the genera. Are the square brackets and dashes necessary?

57-59 unclear, please rephrase. if necessary, divide to two sentences to make it clearer.

117 fresh pork

120 Solarbio – add city, and country of origin

124 Bacteria cultivation and collection

125 cultivation

127 remove the sentence ‘Below describes the procedure to count the number of bacteria in the collected liquid’

128-140 write in full sentences. it seems like you tried to summarize the process but it turned out very confusing. please rephrase.

146 specify how much of the liquid did you take and use for further analysis

203, 375 rephrase small amount ventilation. I understand what are you emphasizing but this is not grammatically correct.

407, 408 do not italicize the explanations in brackets

Reviewer 2 Report

The authors have modified the manuscript almost appropriately compared to previous one.

But there are also many typos, please correct them appropriately.
